# Natural Killer T and Natural Killer Cell-Based Immunotherapy Strategies Targeting Cancer

**DOI:** 10.3390/biom13020348

**Published:** 2023-02-10

**Authors:** Tomonori Iyoda, Satoru Yamasaki, Shogo Ueda, Kanako Shimizu, Shin-ichiro Fujii

**Affiliations:** 1Laboratory for Immunotherapy, RIKEN Center for Integrative Medical Sciences, Yokohama 230-0045, Japan; 2RIKEN Program for Drug Discovery and Medical Technology Platforms, Yokohama 230-0045, Japan

**Keywords:** NKT cell, NK cell, dendritic cells, antitumor effect, cancer immunotherapy, new modality

## Abstract

Both natural killer T (NKT) and natural killer (NK) cells are innate cytotoxic lymphoid cells that produce inflammatory cytokines and chemokines, and their role in the innate immune response to tumors and microorganisms has been investigated. Especially, emerging evidence has revealed their status and function in the tumor microenvironment (TME) of tumor cells. Some bacteria producing NKT cell ligands have been identified to exert antitumor effects, even in the TME. By contrast, tumor-derived lipids or metabolites may reportedly suppress NKT and NK cells in situ. Since NKT and NK cells recognize stress-inducible molecules or inhibitory molecules on cancer cells, their status or function depends on the balance between inhibitory and activating receptor signals. As a recent strategy in cancer immunotherapy, the mobilization or restoration of endogenous NKT or NK cells by novel vaccines or therapies has become a focus of research. As a new biological evidence, after activation, effector memory-type NKT cells lasted in tumor-bearing models, and NK cell-based immune checkpoint inhibition potentiated the enhancement of NK cell cytotoxicity against cancer cells in preclinical and clinical trials. Furthermore, several new modalities based on the characteristics of NKT and NK cells, including artificial adjuvant vector cells, chimeric antigen receptor-expressing NK or NKT cell therapy, or their combination with immune checkpoint blockade have been developed. This review examines challenges and future directions for improving these therapies.

## 1. Introduction

Unlike traditional cancer treatments, such as radiotherapy and chemotherapy, immunotherapy is an innovative treatment method that dynamically modulates the immune system to attack various target cancer cells at multiple locations. It is mainly used to strengthen the immune system by regulating the immune microenvironment, so that immune cells can attack and clear tumor cells at several important locations [1]. With the progress in research on antitumor immunotherapeutic drugs, including immune checkpoint inhibitors (ICIs) and chimeric antigen receptor (CAR)-engineered T cells, the potential market and application of new immunotherapy drugs will increase. 

At present, in addition to conventional antitumor therapy and recently emerging ICIs, tumor immunotherapy is being developed from different aspects, such as innate immunity (natural killer T (NKT) and natural killer (NK) cell activation) and adaptive immunity (CD4^+^ and CD8^+^ T cell activation). After stimulating immune cells with a therapeutic vaccine, the body’s own tumor-specific immune response can block a tumor’s ability to escape immune surveillance. Immune cells can thus play a role in tumor surveillance and tumor cell clearance. In contrast, adoptive cell immunotherapy (ACI) is effective against hematologic tumors, but due to the heterogeneity within solid tumors and the local microenvironment, the efficacy of ACI against solid tumors may be limited. ACI treatment involves injecting immunologic effector cells expressing modified and amplified genes to kill tumor cells. ACI can be specific or nonspecific. Although nonspecific T cells, such as nonspecific ACI, are classified as activated lymphocytes, they exhibit only a weak tumor-killing activity. Specific ACI is induced by specific tumor–antigen stimulation and includes tumor-infiltrating lymphocyte therapy, T-cell receptor-T (TCR-T), and CAR T-cell immunotherapy (CAR-T) [2]. The main effector cells are CD8^+^ T and CD4^+^ T cells. This therapy can have strong specificity and targeting ability; hence, it can be used in patients with advanced cancers or patients showing no response to other therapies. 

The innate immune system is the first line of defense against infection or cancer, and it takes time to induce an antigen-specific T cell response. When the body detects pathogens or cancer cells, it activates the innate immune system to attack and destroy them, while informing and modulating the adaptive immune response. As conserved innate effector lymphocytes, NK, NKT, γδ T, and mucosa-associated invariant (MAIT) cells are crucial for immune surveillance [3,4]. Notably, antigen recognition by the aforementioned unconventional T cells is not restricted to MHC class I and II molecules [3]. NKT cells are characterized as type I or type II [5,6]. Both types recognize glycolipids on CD1d molecules, but their functions in cancer defense are clearly distinct [7]. In this review, we focus on type I NKT cells, also known as semi-invariant NKT (iNKT) cells, which exhibit an adjuvant activity by linking innate and adaptive immunity through in situ dendritic cells (DCs) [5]. Especially, we highlight cancer immunotherapy utilizing the properties of iNKT cells, such as cytotoxicity, cytokine production, and tissue migration.

Next, we discuss how NK cells, which are a major component of the innate immune system, play a pivotal role in cancer immune surveillance. NK cells usually express killer inhibitory receptors (KIRs) and killer activation receptors. They recognize normal cells via KIRs in a dominant manner and avoid self-killing. When activated after the recognition of target cells, NK cells eliminate a variety of abnormal or stressed cells, including tumor and infected cells [8]. The effect of NK cell cytotoxicity on tumors, especially metastatic and hematologic tumors, can be shown in MHC-I-deficient tumor cells or tumor cells with upregulated expression of activated ligands [8]. Furthermore, activated cytokines, such as IL-2 and IL-15, mediate and promote the activation of NK cells, and antibody-dependent cell-mediated cytotoxicity (ADCC) mediates NK cell’s ability to recognize and kill antibody-coated tumor cells. In addition, the recently developed bispecific T cell-engager approach can improve the effectiveness of ADCC [9]. 

Taken together, in this review, we discuss the biological bases of iNKT and NK cells, results of preclinical studies and clinical trials, and prospects for including iNKT and NK cells in the arsenal of cancer immunotherapies.

## 2. Characterization of iNKT Cells and Anti-Cancer Immunotherapy

iNKT cells from mice express an invariant TCR, which is formed by the rearrangement of the Vα14 and Jα18 TCRα gene segments. iNKT cells are preferentially paired with Vβ chains, including Vβ8.2, Vβ7, or Vβ2 TCR β gene segments [10,11,12]. In humans, iNKT cells express a rearranged Vα24-Jα18 TCRα-chain associated with a Vβ11 TCRβ-chain [13,14]. When iNKT cells are stimulated by a ligand, such as α-galactosylceramide (α-GalCer), they can produce large amounts of cytokines, such as IFN-γ, TNF-α, IL-2, and IL-4 [15,16]. Murine iNKT cells can be categorized into at least three distinct functional subsets, iNKT1, iNKT2, and iNKT17 cells, and this classification is regulated by the transcription factors T-bet, GATA3, and RORγ, respectively [17,18]. Each iNKT cell subset produces typical cytokines upon activation (IFN-γ, IL-4, or IL-17). A small population of iNKT17 cells is located in the lung and subcapsular regions of lymph nodes (LNs) [17]. Human iNKT cells develop within the thymus in a PLZF-dependent manner, similar to murine iNKT cells [19,20]. However, subsets of human iNKT cells are not as fully defined as those of mouse iNKT cells. Among CD4^+^ iNKT cells, CD8^+^ iNKT cells, and double-negative (DN) iNKT cells, DN iNKT and CD8^+^ iNKT cells are similar to mouse iNKT1 cells, with increased IFN-γ secretion and cytotoxic function when activated [21,22]. CD4^+^ iNKT cells produce a relatively higher amount of Th2-type cytokines, such as IL-4 and IL-13, than other subsets [21,22]. Human iNKT cells, particularly CD4^-^ iNKT cells, predominantly express certain NK-related markers, such as 2B4, NKG2D, DNAM-1, CD94, and NKG2A [23]. The cytotoxicity of human iNKT cells against target cells may be mediated by either TCR or natural cytotoxicity receptor-mediated signaling. 

iNKT cells recognize and kill tumor cells via NKG2D, CD161, NKp44, and NKp80 as well as via endogenous NKT ligands loaded on the CD1d molecule. Conversely, tumor cells suppress iNKT cell function by expressing PD-L1, FGL-1, galectin-9, and HVEM [24,25,26,27]. Further, the tumor microenvironment is inhibitory to immune cells, and lactic acid inhibits IFN-γ production by blocking mTORC1 in iNKT cells and suppressing PPARγ-mediated cholesterol synthesis [28] (Figure 1).

Owing to their peripheral differentiation, iNKT cells can differentiate into memory-type iNKT cells. iNKT cells recognize a variety of lipid antigens, including ceramide-based glycolipids, such as glycosphingolipids, microbial lipids, and endogenous self-lipids, presented by MHC class I-like CD1d molecules on antigen-presenting cells (APCs). These lipid antigens are either intermediates of APCs’ intracellular metabolism or originate from the cell walls of bacteria, fungi, or protozoan parasites. 

Various α-linked ceramide glycolipids are antigenic glycolipids produced only by microorganisms. α-glucuronosylceramides and α-galacturonosylceramides are produced by *Sphingomonas spp*., α-galactosyldiacylglycerols are produced by *Borrelia burgdorferi*, and α-glucosyldiacylglycerols are produced by *Streptococcus pneumoniae* [29]. α-GalCer, a prototypical iNKT cell ligand, is a sphingolipid that was initially isolated from the marine sponge, *Agelas mauritiana*, in 1994 [30]. Although some human cells produce β-GalCer via the glycosphingolipid metabolic pathway, there is no direct pathway in human cells to produce α-GalCer. Recently, it was reported that α-GalCer, rather than β-GlcCer or β-GalCer, in mouse gut was affected by the presence of the antibiotic vancomycin and not colistin [31]. Human gut bacteria, such as *Bacteroides fragilis*, *Bacteroides vulgatus*, and *Prevotella copri*, produce α-GalCer structures that can activate iNKT cells. α-GalCer can also be detected in certain tumor tissues, such as colon adenocarcinoma tissues. In fact, the presence of *B. vulgatus* in colon adenocarcinoma tissues correlates with better survival. Thus, α-GalCer-producing bacteria, members of the human gut microbiome, may infiltrate tumor tissues, resulting in antitumor activities [31]. Endogenous CD1d-binding iNKT cell ligand glycolipids, such as isoglobotrihexosylceramide (iGb3) [32,33], disialoganglioside GD3 [34], and α-glycosylceramides, have also been identified [35,36].

Tumor-derived lipids can help tumors evade immune surveillance using several mechanisms. An alteration in tumor lipid profile or the accumulation of specific lipids and fatty acids that favor tumor growth can limit antitumor immunity [37]. Sphingosine-1-phosphate (S1P) expression is upregulated in mantle cell lymphoma (MCL), an aggressive subtype of non-Hodgkin lymphoma. Knockdown of sphingosine kinase 1, the enzyme responsible for producing S1P, in human MCL cells induces upregulation of the expression of cardiolipin and glycerophospholipid, which are abundant in mitochondrial membranes and bind to CD1d [38], increasing iNKT cell activation [39]. Another report showed that iNKT cells from the infusion of donor lymphocytes efficiently lysed leukemia cell lines and primary acute myeloid leukemia (AML) blasts in a dose- and CD1d-dependent manner [40]. This implies that tumor metabolites present on tumor cell CD1d may be potential targets of iNKT-mediated immunotherapy against hematologic malignancies. Although CD1d expression varies among tumor types, in B cell chronic lymphocytic leukemia (CLL), the most common hematologic malignancy, the expression of CD1d is lower than that in normal B cells or is absent [41], which might allow the tumor to escape from iNKT-mediated surveillance.

iNKT cells are part of the innate immune system, and effector memory-like iNKT cells have also been shown to function as other innate immune cells, such as NK and γδT cells, which differentiate into memory cells after activation. Human iNKT cells may mediate immune response against *Mycobacterium tuberculosis*. In humans, CD3^+^TCR Vβ11^+^ NKT cells from pleural fluid mononuclear cells exhibit an effector memory phenotype. Increased levels of expression of cytolytic molecules and *M. tuberculosis* antigens stimulate the production of IFN-γ, indicating that iNKT cells participate in local immune responses against *M. tuberculosis* and protection against *M. tuberculosis* infection [42]. Killer cell lectin-like receptor subfamily G member 1-positive (KLRG1)-expressing iNKT cells were induced in the lungs after vaccination with α-GalCer-loaded CD1d^+^ cells, including α-GalCer-loaded DCs or α-GalCer-loaded CD1d-expressing NIH-3T3 cells. KLRG1^+^ iNKT cells co-expressing CD49d and granzyme A persisted for several months and retained a potent response to secondary stimulation [43]. Shimizu et al. also revealed that Eomes plays two roles in iNKT cells. First, during iNKT cell development in the thymus, Eomes regulates iNKT1 cell differentiation. Second, Eomes promotes differentiation into effector memory cells in peripheral organs and spleen. Eomes-conditional deficient iNKT cells have been shown to fail to differentiate into KLRG1^+^ granzyme A^+^ long-term effector memory cells, even after stimulation with α-GalCer-loaded DCs [44]. Furthermore, Prasit et al. reported that upon intratumoral administration of a TLR9 agonist, CpG and α-GalCer suppressed tumor growth in murine tumor models. This treatment also suppressed distant untreated tumors, known as the abscopal effect. They showed that sustained activity of iNKT cells was associated with the infiltration of KLRG1^+^ effector memory iNKT cells into both tumors and regional LNs [45].

## 3. Immunological and Clinical Findings of iNKT-Based Immunotherapy

Various therapies have been developed that are effective for treating hematopoietic tumors. Activating iNKT cells is an attractive strategy for anti-cancer therapy for several reasons. First, iNKT cells are available for treating many types of cancers because all human beings share the same invariant TCR and CD1d [5,46]. Second, iNKT cells can not only kill cancer cells directly but also induce activation of NK cells as an adjunct effect [47]. Third, iNKT cells exert an adjuvant effect on T cells via DCs. Fourth, iNKT cells can alter the tumor microenvironment by inhibiting immunosuppressive cells, such as tumor-associated macrophages and myeloid-derived suppressor cells [48,49]. Although iNKT cell level is often decreased in patients with certain cancers, whether iNKT cells can be expanded or not is of higher importance than their level in patients [50]. Reduced levels and functional impairment of iNKT cells have been reported in a wide range of solid and hematologic malignancies, including prostate [51], lung, esophageal, colorectal, gastric, gallbladder, uterine, bile duct, and pancreatic cancers [52] and multiple myeloma [53] as well as in patients with late stage head and neck squamous cell carcinoma (HNSCC) [54], neuroblastomas [55], AML, and CLL [56]. Conversely, increased level of intratumoral iNKT cells is correlated with good clinical outcomes and improved survival in colorectal cancer, neuroblastoma, periampullary adenocarcinoma, and hematologic malignancies [57]. In prostate cancer and oral cell squamous carcinoma, iNKT cells exhibit defective IFN-γ production but acquire IL-4 production ability [51]. Studies on cancer patients also showed that iNKT cells respond to chemotactic signals released by tumor cells, particularly in a range of primary and metastatic solid tumors. iNKT cell infiltration is associated with CCL2 expression in neuroblastoma cells [58] and CCL20-producing tumor-associated macrophages [59]. Patient-derived iNKT cells can be restored in vitro by culturing them in the presence of IL-2 or IL-12 or in vivo upon therapeutic administration of APCs pre-loaded with α-GalCer. Therefore, several clinical trials using autologous α-GalCer-pulsed DCs or adoptive transfer of ex vivo expanded iNKT cells have been conducted. α-GarCer-pulsed DC injections were well tolerated in all patients, with no major toxicity or altered immune response. The best clinical responses included decreased urine or serum M protein level in three cases of myeloma, one stable disease in a patient with renal cell cancer, and a reduction in tumor-associated monoclonal immunoglobulin level by administering α-GarCer-pulsed DC in combination with low-dose lenalidomide in three of four patients with measurable disease [60]. In patients with advanced lung cancer, IFN-γ-producing iNKT cell therapy, including Vα24^+^ iNKT cell and α-GalCer-pulsed APC (APC/Gal) therapy, correlated with clinical responses in 17 patients with non-small cell lung cancer [61]. We established an mRNA-transfected cell-based vaccine (the artificial adjuvant vector cell (aAVC) system [62,63,64]) that was co-transfected with an antigen-derived mRNA and CD1d mRNA, and subsequently loaded with iNKT cell ligand. When administered intravenously, aAVC activated iNKT and NK cells, and the activated iNKT and NK cells rejected aAVC. Subsequently, the killed aAVCs were taken up by DCs in situ, thereby activating several DC-specific immunogenic features. Previous studies on tumor therapeutic models reported that administration of tumor antigen (e.g., melanoma-associated antigen recognized by T cells-1 and Wilms’ tumor 1 (WT1))-expressing aAVCs induced potent CD4^+^ and CD8^+^ T cell responses [63,64,65]. Recently, we developed WT1-antigen-expressing aAVC (aAVC-WT1) therapy for relapsed and refractory AML patients [66] and completed the first human clinical trial of aAVC-WT1 therapy (Table 1).

Clinical studies involving adoptive transfer of iNKT cells have also been performed (Table 1). Ten patients with locally recurrent and operable HNSCC were treated with ex vivo expanded APC/Gal. Half of the patients achieved objective tumor regression [67]. In contrast, ex vivo expanded iNKT cell transfer studies showed an increase in iNKT cell levels in the peripheral blood (PB), and in all cases, the number of IFN-γ-producing cells in PB mononuclear cells (PBMCs) increased. Despite an immunological response, no clinical response was observed [68]. In another study, when iNKT cells expanded using anti-CD3 mAb and IL-2 were infused into patients three times at 2 weeks interval, nine patients exhibited grade 1–2 toxicities, while three patients showed no evidence of disease or stable disease [69].

Next generation engineered iNKT cell transfer approaches have already been introduced (Figure 2). Engineered human hematopoietic stem cells (HSCs) expressing a human iNKT TCR gene were examined for their ability to produce human iNKT cells that can target cancer. These HSC-derived iNKT cells effectively suppressed multiple myeloma cells in a xenograft model in a CD1d-dependent manner [76]. Using genetically engineered CD34^+^ HSCs, yield and purity of human allogeneic CD34^+^ HSC-engineered iNKT (^Allo^HSC-iNKT) cells have been increased to increase the number of endogenous iNKT cells. ^Allo^HSC-iNKT cells exhibited enhanced tumor-killing efficacy against five tumor cell lines due to increased expression of NK-activating receptors and increased production of highly cytotoxic molecules. BCMA-targeting CAR-engineered ^Allo^HSC-iNKT cells targeting MM tumors using the NK/TCR/CAR triple mechanism exhibited increased cell-killing activity [77]. Furthermore, third-party HSC-iNKT cells are useful as preclinical models of lymphoma and leukemia [78]. Moreover, engineered NKT cells co-expressing both anti-GD2 CAR and IL-15 were used in a phase I study for children with relapsed or resistant neuroblastoma. The results showed that CAR-NKT cells were expanded in vivo and migrated to the tumor site, resulting in the regression of bone metastatic lesions in one patient [70]. 

## 4. Characterization of NK Cells and Anti-Cancer Immunotherapy

NK cells were initially identified as lymphocytes capable of killing tumor cells or virally infected cells with effector functions but are now recognized as a subset of group 1 innate lymphoid cells (ILCs) [79]. ILC family members, including ILC1, ILC2, and ILC3, are derived from the same common lymphoid progenitors (CLPs) in the bone marrow as those of T and B cells [79]. In mice, CLPs develop into pre-NK precursors (pre-NKPs) and then transition into refined NKPs (rNKPs), expressing the IL-2 receptor β chain (CD122), which enables them to respond to IL-15 [79,80]. Subsequent expression of NKG2D followed by NK1.1 occurs when the progenitors reach the immature NK (iNK) stage [80,81,82]. The transcription factors, RUNX3, CBF-b, STAT5, NFIL3, PU.1, Notch, and TCF-1, are critical for the development of pre-NKPs and rNKPs, whereas T-bet, Eomes, and Zeb2 have been widely reported to play an important role in NK cell maturation [83]. NK cells are different from ILC1, owing to the fact NK cells express Eomes and exhibit cytotoxic activity [79,82]. Human NK cells are derived from multipotent CD34^+^ hematopoietic progenitors in the bone marrow [80,81]. CD34^+^CD45RA^+^ NK cells are stage 1 NK cells and are also located in the bone marrow, while stage 2–4 NK cells are found in secondary lymphoid tissues, such as tonsils, LNs, and spleen. Development into stage 2 (acquisition of CD56, CD117, and IL-1R1), stage 3 (loss of CD34), stage 4 (downregulation of CD117 expression with acquisition of CD94; CD56^bright^ NK cells), stage 5 (CD56^dim^ NK cells), and stage 6 (CD57^+^ NK cells) is observed in the bone marrow as well as in lymphoid organs [81]. Recent studies using high dimensional analysis and single-cell RNA transcriptome analysis revealed that NK cells are highly heterogeneous with diverse functions, but two main subsets of NK cells are conserved in humans and mice: NK1 (CD56^dim^ NK in humans and CD27^-^CD11b^+^ NK in mice) and NK2 (CD56^bright^ NK in humans and CD27^+^CD11b^−^ NK in mice) [84,85,86]. CD56^dim^CD16^+^ NK cells were predominantly located in the blood, bone marrow, spleen, and lung, whereas CD56^bright^ CD16^-^ NK cells were enriched in the LNs, tonsils, and gut [87]. Tissue-resident NK (tr-NK) cells were recently observed to be distinct from circulating NK cells [88,89,90]. In fact, they express CD69 and CD103, similar to tissue-resident memory T cells [89,90]. tr-NK cells in the liver are CD56^bright^ NK cells that express distinct chemokine receptors, namely CXCR6, CXCR3, and CCR5, whereas decidua tr-NK cells are CD56^bright^CD16^-^KIR^+^CD9^+^CD49a^+^ cells [89,90]. tr-NK cells in the bone marrow have a CD56^bright^CD44^+^CD62L^+^ CXCR3^+^ CD52^+^CD160^-^ phenotype [87,90].

NK cells exhibit potent cytolytic activity to kill targeted cells and simultaneously secrete various inflammatory cytokines (IFN-γ, TNF-α, G-CSF, and GM-CSF) and chemokines (CCL3, CCL5, XCL1/2, and FLT3LG) [90,91,92]. In contrast to T and B cells, NK cells utilize an array of germline encoding activating and inhibitory receptors to recognize target cells, such as infected, stressed, or cancer cells (Figure 3) [92].

The activation status of NK cells depends on the balance between inhibitory and activating signals from receptors. NK cell-activating receptors include NCRs (NKp30, NKp44, and NKp46), c-type lectin (NKG2D), DNAX accessory molecule 1 (DNAM-1), Fc receptors (CD16), and killer cell immunoglobulin-like receptors (KIRs) [80,82,90]. NK cell inhibitory receptors include Ly49-type inhibitory receptors in mice and KIRs, KIR2DL, KIR3DL, and the CD94-NKG2A heterodimer in humans (Figure 3) [93]. These receptor signals are triggered by “missing-self” and “induced-self-ligand” interactions [92]. The “missing-self” recognition occurs when the relative expression of NK-inhibitory MHC-I molecules exceeds the expression of NK cell-activating ligands, thereby leaving NK cells inactive. In contrast, increased expression of “induced-self ligands” on malignant cells coupled with reduced expression of MHC-I leads to the activation of NK cells exhibiting potent cytolytic activity. CD56^dim^CD16^+^ NK cells selectively express KIRs, whereas CD56^bright^CD16- NK cells express CD94/NKG2A (Figure 3) [92,93,94]. When activated, NK cells kill target cells via cytotoxic proteins, including perforin, granzyme family members, and death receptor signaling proteins (Fas ligand and TNF-related apoptosis-inducing ligand) [92].

NK cells also play a role in the pathogenesis of hematologic malignancies. It has been reported that NK cell levels are significantly reduced and have impaired cytotoxic function and development in patients with hematologic malignancies, such as AML and MM [86,95,96]. In AML, the expression levels of activating receptors, such as NKG2D, NKp46, and NKp30, on NK cells are downregulated. AML blasts activate the AHR pathway in NK cells by producing AHR agonists, and upregulation of miR-29b expression in NK cells impairs their maturation. In addition, upregulation of GSK3-β expression in NK cells has been reported. Inhibition of GSK3 activates NK cells to produce TNF-α and upregulates LFA-1 expression, leading to synapse formation with AML and killing of AML blasts [95,97]. In MM, there are several inhibitory mechanisms: Treg- and TAM-derived TGF-β, PGE2, or IL-15/IL-15R axis inhibition via BM stroma cells, and HLA-E expression on MM cells impair NK cell function [81]. 

## 5. Immunological and Clinical Findings of NK-Based Immunotherapy

NK cell-based immunotherapy is currently a promising therapeutic approach for cancer (Figure 2 and Table 1) [82,98,99,100]. There are several sources of NK cells, including cells derived from PB, umbilical cord blood, NK cell lines, and HSC/iPSCs cells, and several types of genetic engineering protocols can be used to generate NK cells. Cord blood (CB)-derived NK cells from blood banks are a promising source for NK cell-based therapy. CB-NK cells primarily contain immature NK cells that, after expansion, acquire a cytotoxic status functionally equivalent to that of PB-NK cells. NK cells isolated from adult PBMCs are believed to be healthy and non-exhausted NK cells that have relatively low cytolytic capacities. As they are required to prime NK cells, ex vivo expansion of NK cells by supplementation with IL-2, IL-15, and IL-21, to generate cytokine-induced memory-like NK (CIML-NK) cells, has been investigated. CIML-NK cells show increased cytokine production, cytotoxic function, and proliferation rates [101]. In a clinical trial (NCT03068819), nine children/adolescents with relapsed AML after hematopoietic stem cell transplantation were treated with donor-derived CIML-NK cells. Four of the 8 evaluable patients achieved complete remission on day 28. Two patients maintained durable remission for >3 months, with one patient in remission for >2 years [71].

Several strategies for off-the shelf NK cell-based immunotherapy have been developed. Initially, the NK lymphoma-derived cell line NK-92 was evaluated as an Investigational New Drug approved by the US Food and Drug Administration. NK-92 cells have the advantage of easy ex vivo expansion for clinical use and lack most KIRs [102]. Nevertheless, NK-92 cells have several limitations, including low ADCC, due to low or absent CD16 expression, and requirement for irradiation prior to infusion. To date, clinical safety has been reported, but no sustained responses have been reported. Furthermore, NK cells, FT500, FT516, and FT538, have been derived from a clonal iPSC master line. FT516 is designed to express the novel, high-affinity, and non-cleavable valine 158 polymorphism of the CD16 (hnCD16) Fc receptor, which has been modified to prevent downregulation of its expression and enhance its binding to tumor-targeting antibodies. FT516 has been investigated in two phase 1 clinical trials (NCT04023071 and NCT04551885). FT538 promotes survival and function in environments with high oxidative stress. In a multi-dose phase 1 clinical trial, FT538 was used for the treatment of AML, in combination with daratumumab, a CD38-targeted monoclonal antibody therapy used for the treatment of multiple myeloma (NCT04614636). Another type of NK cell therapy, CAR-NK cell therapy, has also been the focus of research because the design of CAR molecules is suitable for use with NK cells. The transmembrane domains of CAR-T are usually unitized CD3z, CD8, and CD28, but in the case of NK, molecules normally found on NK cells, such as DNAM-1, 2B4, and NKG2D, are superior in terms of cell proliferation, cytokine production, and antitumor activity. In clinical trials, CD19, BCMA, CD33, and CD22 were targeted for hematologic tumor treatment [93]. The first CAR-NK clinical trial (NCT02944162) included only three patients with relapsed or refractory AML and used anti-CD33 CAR NK-92 cells as therapeutic agents; however, this study did not show durable responses [72]. Another CD19-CAR-NK cell line, using NK cells derived from CB, was administered to 11 patients with non-Hodgkin lymphoma or CLL (NCT03056339). This clinical trial reported that seven out of 11 patients had complete remission without the development of major toxic effects, and the infused CAR-NK cells expanded and persisted at low levels for at least 12 months [73]. FT596, an iPSC-derived, off-the-shelf cell line, was used to generate CD19-directed CAR-NK cells (Fate Therapeutics). Interim analysis by a phase 1 clinical trial using FT596 in combination with an anti-CD20 antibody for treating R/R BCL (NCT04245722) revealed an estimated overall response rate of 69% and a complete response of 56% [74]. Thus, CAR NK cells exhibit remarkable cytotoxicity against hematologic malignancies. To improve efficacy and suppress adverse events, several engineered CAR NK cells have been developed, including NKG2D-CAR, which expresses NKG2D-4-1BB-CD3z-CAR, and dual CAR, which co-expresses anti-CD19 or CD33-activating CAR and anti-HLA-DR iCAR [103,104]. Similar to combination therapy of CAR-T cell therapy [105], CAR-NK cells can be combined with radiotherapy, which induces DNA damage in tumor cells, followed by upregulating NKG2D ligands on tumor cells, NK cell cytotoxicity against cancer cells was promoted [106]. 

Other modalities, using bi-specific and tri-specific NK cell engagers or immune checkpoint blockade, have been used for NK cell therapies. First, several T cell-engager technologies are currently being used to develop NK cell engagers. NK cell-engager strategies include tri-specific and tetra-specific designs that strengthen the antitumor effect by targeting multiple antigens on the tumor or by cross-linking cytokine moieties to support NK cell expansion and survival [107]. Antibody Fc domains have been engineered by introducing point mutations or modifications (for low-fucose antibodies), with the aim of increasing their ability to bind to FcγR. CD16A, a member of the immunoglobulin superfamily, serves as a receptor for the Fc fragment of IgG. In addition to CD16A, NKG2D, CD94/NKG2C, NKp30, and NKp46 are also targeted by NK cell engagers [108,109]. For hematologic malignancies, CD33 and CLEC12A (AML), BCMA, and CD38 (MM), and CD30 (PTCL) are targeted as tumor arms [110,111]. Several NK cell engagers have been used in clinical trials. AFM13, a bi-specific engager-binding CD16 on NK cells, and CD30 on leukemia or lymphoma targets, exhibit enhanced killing of CD30^+^ tumor cells, leading to CAR-like responses (NCT03192202). TriKE (161533) comprises two single-chain variable fragments (scFvs), one that binds to CD16 on NK cells and another that binds to CD33 on myeloid malignancies, plus an IL-15 linker bridging the CD16 and CD33 scFvs for sustained cell activation [112].

Anti-CD94/NKG2A (IPH2201-Monalizumab) has been used in various clinical trials as an NK cell-based immune checkpoint blockade. Using humanized anti-NKG2A antibody against hematologic malignancies has proven to be safe and effective [113]. Improvement of NK cell dysfunction by monalizumab in CLL has been demonstrated in vitro [114]. Monalizumab was well tolerated as a monotherapy in gynecological malignancies, with no reported dose-limiting toxicities or serious adverse events. This ongoing trial of heavily pretreated cohorts revealed stable disease in 41% of evaluable patients [75].

## 6. Conclusions

Both NKT and NK cells represent specialized effector cell populations with a fast-acting and potent antitumor capacity. However, there are still some limitations in NKT and NK cell-based immunotherapy. One of them is that they are short-lived. However, since recent studies have focused on long-lived or memory NKT or NK cells, such a strategy would be useful in the future. Next, to harness both as powerful effectors, a strategy is needed to reliably recognize and kill tumor cells. Further, such strategies need to be designed as off-the shelf strategy. For this, several attempts have been made to move toward antigen-specific immunotherapies, including CAR-NKT and CAR-NK cells by transducing CAR or artificial adjuvant vector cells as allogeneic cell-based vaccine modality. With the new tools using genetic engineering or new modalities of cell therapy, iNKT and NK cell-based therapies are emerging as a complementary therapeutic modality to T cells.

## Figures and Tables

**Figure 1 biomolecules-13-00348-f001:**
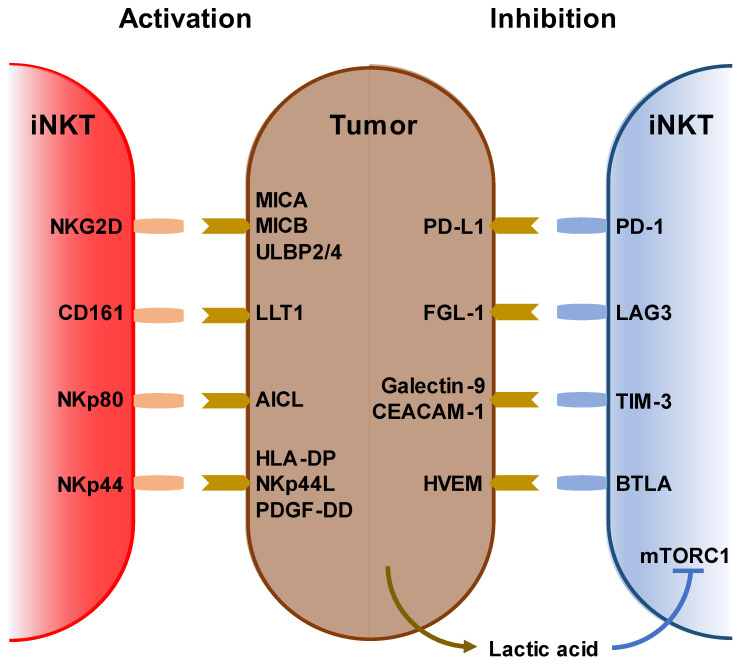
Stimulatory and inhibitory receptors on iNKT cells. (Left) Ligands and receptors for activating iNKT cells. (right) Ligands and receptors for suppressing iNKT cells. Lack of oxygen increases lactic acid in the tumor microenvironment. Lactic acid also inhibits iNKT cell activation via mTORC1 signaling.

**Figure 2 biomolecules-13-00348-f002:**
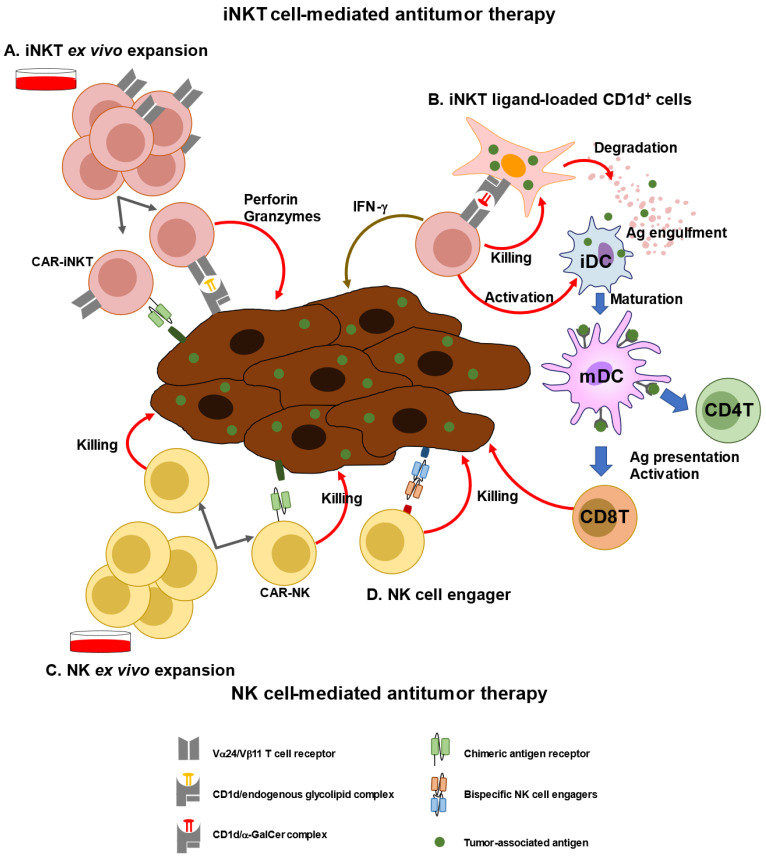
iNKT or NK cell-mediated immunotherapy. (**A**) iNKT cells can be expanded ex vivo using specific ligands (i.e., a-GalCer) and cytokines (e.g., IL-2, and IL-15). Expanded iNKT cells or chimeric antigen receptor (CAR)-transfected NKT cells can be adoptively transferred to patients. (**B**) As a new modality of vaccines, iNKT ligand-loaded CD1d^+^ cells with tumor-associated antigen (TAA), called artificial advent vector cells, can induce TAA-specific CD4^+^ and CD8^+^T cells via maturation of dendritic cells. (**C**) NK cells can be expanded ex vivo using cytokines, such as IL-15 and IL-2. CAR-transfected NK cells can be adoptively transferred to patients. (**D**) Bispecific NK cells exhibit cytotoxic activity against tumor cells by promoting cell–cell interaction.

**Figure 3 biomolecules-13-00348-f003:**
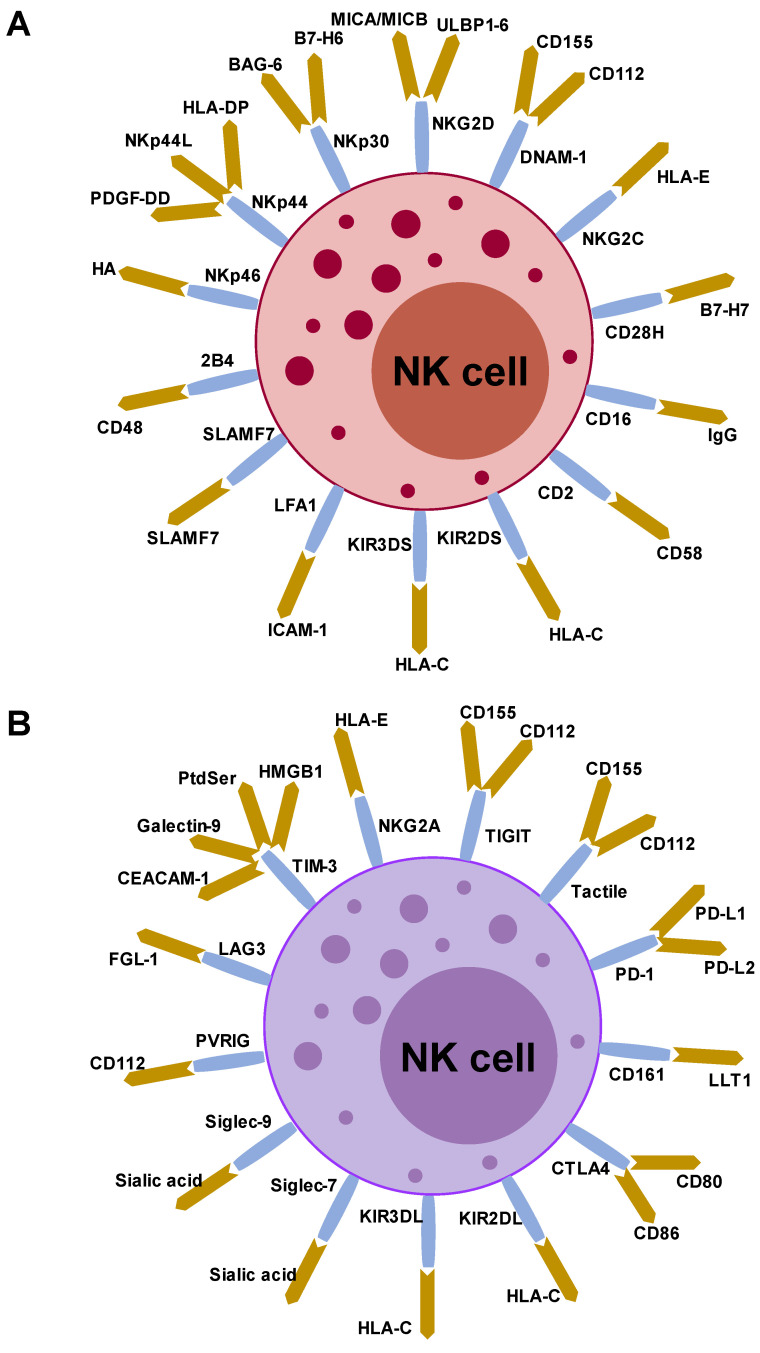
Stimulatory and inhibitory receptors on human NK cells and their ligands. (**A**) NK cells express various activating and inhibitory receptors. Stimulatory receptors on NK cells and their ligands. (**B**) Inhibitory receptors on NK cells and their ligands. HA, hemagglutinin; PtdSer, phosphatidylserine.

**Table 1 biomolecules-13-00348-t001:** iNKT and NK cell-mediated immunotherapy.

Target	Modalities	Cancer	Phase	Mechanism	Reference or Clinical Trial Number
iNKT	α-GalCer-pulsed dendritic cells or antigen presenting cells	myeloma, lung	I	Activation of iNKT and NK cells	[60,61]
	Artificial adjuvant vector cell	AML	I	Both NK and CTL were generated DC maturation in situ	[66]
	Ex vivo expanded iNKT cells	HNSCC, lung, melanoma	I	Direct cytotoxicity by expanded iNKT cells	[67,68,69]
	CAR-NKT	Neuroblastoma	I	NKT-mediated cytotoxicity against GD2^+^ cancer cells	[70]
NK	Ex vivo expanded NK cells	AML	I	Direct cytotoxicity by expanded iNKT cells	[71]
	iPSC-derived NK cells	AML, B-cell lymphoma, multiple myeloma	I	NK cells are regenerated from a clonal iPSC master cell line.	NCT04023071, NCT04551885, NCT04614636
	CAR-NK	AML, non-Hodgkin lymphoma, CLL, BCL	I	Direct cytotoxicity by anti-CD19/CD33 CAR expressing NK cells	[72,73,74]
	NK cell engager	Leukemia, lymphoma	I/II	Binding CD16 on NK cells and CD30 on leukemia or lymphoma	NCT03192202
	Immune checkpoint blockade	Ovarian carcinoma, squamous cervical carcinoma, END	II	Block the binding between CD94/NKG2A and HLA-E	[75]

iNKT: invariant natural killer T cells, NK: natural killer cells, AML: acute myeloid leukemia, HNSCC: head and neck squamous cell cancer, Ab: antibody, iPSC: induced pluripotent stem cells, CAR: chimeric antigen receptor, BCL: B-cell lymphoma, CLL: chronic lymphocyte leukemia, END: epithelial endometrial carcinoma.

## Data Availability

Not applicable.

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
