# Peer review of "Natural Killer T and Natural Killer Cell-Based Immunotherapy Strategies Targeting Cancer"

_biomolecules, 2023, doi:10.3390/biom13020348_

Round 1

Reviewer 1 Report

Comments on biomolecules-2127331

In this study, the author has studied “NKT and NK cell-based immunotherapy strategies targeting cancer.” A lot of studies have already been carried out on a similar topic, and comprehensive data is available in the literature. The English language used in the manuscript needs major improvements as there are some punctuation and grammatical mistakes present throughout the manuscript. Experimental designs required more clarity. Moreover, research models are not discussed in an understandable manner. Repetition of lines is common, reflecting that the author needs a more comprehensive way of thinking. Important mechanisms are missing.

Specific comments:

1. Please revise the title. It is recommended to use the full term instead of abbreviations.

2. The Abstract needs to be critically revised, please limit the background knowledge to a few sentences and add more results. 

3. Please add more strong keywords.

4. Page 1, line 34-35: “With the progress in antitumor immunotherapeutic drug research...” Please enlist a few drugs.

5. On which bases immunotherapy is considered to outperform the other therapies?

6. Page 2, line 54-56: “The innate immune system is the first line of defense… the adaptive immune response.” Please reference here and also add more data. The average length of the paragraph should be 10-15 lines.

7. Page 2: The whole introduction section looks general. Authors are advised to revise the introduction section carefully and add relevant data to support the problem statement and make a connection between each paragraph. There is no such information between NKT/NK and cancer rather than general information on NKT/NK and its types. Please add relevant literature to support the problem statement. Overall, an introduction needs a major revision.

8. Page 2: What is the research gap and novelty of the present study?

9. Fig. 1 is very general; these types of figures are already present in the literature. Please use any specific figure to illustrate the mechanism of iNKT.

10. Page 4, line 183-184: “Although iNKT cell level is often decreased in patients with certain cancers…” Which certain cancers?

11. Please add more mechanisms of action (figures) NKT/NK against cancer.

12. It is recommended at a table of clinical trials.

13. Adding a detailed table at the end with different cancer aspects explaining the iNKT/NK immune therapy with a mechanism is recommended.

14. Authors are advised to proofread the manuscript to overcome grammatical mistakes.

15. It is recommended to add some data on immune evasion and immunosuppression, iNKT/NK receptors and their ligands in humans, infiltration of NK, and present challenges and future directions.

16. Authors are advised to revise headings and subheadings.

17. Most of the references are outdated; please revise them and add updated data. 

Reviewer 2 Report

The perspective of the review writing on “NKT and NK cell-based immunotherapy strategies targeting cancer” is notable and has insightful knowledge. This review is well-structured having an interesting context about iNKT and NK cells and their immunotherapeutic role.

I have a few minor suggestions for the author, please clarify and corrected these errors.

1.    In line number 86-89, the sentence “Murine iNKT cells can be categorized into at least three distinct functional subsets, iNKT1, iNKT2, and iNKT17 cells, and this classification is regulated by the transcription factors T-bet, GATA3, PLZF, and RORγ, respectively” is not correct. The author should rewrite this sentence more explicitly and specifically for each subset of iNKT cells.

2.    Author should write Double Negative (DN) for DN iNKT cells.

3.    In figure legend 2, the author has defined the various activating and inhibitory receptors and their ligands but forgot to explain that these receptors belong to mouse or human NK cell receptors.

4.    Author should correct reference number 94.

5.    Author should maintain uniformity in the references such as page number and use of all authors’ names in the references.

6.    Minor typological errors are noticed in the manuscripts please correct them before final submission.

7.    Please check the grammar mistakes and spelling check. In some places, the hyphen is missing.

8.    Author efforts are observed, to input a lot of information presented in a summarized manner. This review has very impressive writing skills and is well-connected. Overall, the manuscript is informative, nice, and ready to publish.  

Reviewer 3 Report

Dear Author,

This is an interesting review but I expect it should be revised with a few good images and add clinical trials and combination with other important strategies. Also, I think future prospects for clinical translation are missing, and it's just 10 page review with no single table for critical analysis.

Author Response

  1. This is an interesting review but I expect it should be revised with a few good images and add clinical trials and combination with other important strategies.

----Thank you for the comment. We have added on page 8, lines 396–399 information on CAR-NK in combination with other therapies.

  1. Also, I think future prospects for clinical translation are missing, and it's just 10 page review with no single table for critical analysis.

----As suggested, we have included a new table.

Reviewer 4 Report

Authors have presented a manuscript entitled “NKT and NK cell-based immunotherapy strategies targeting cancer”. Authors have selected an emerging topic and highlighted the clinical aspects of iNK and Nk cells in immunotherapy-based cancer treatment. However, manuscript still needs some major corrections before publication.

·       First of all, I would like to suggest that authors should aligned the content of manuscript as per the abstract section. In abstract section authors have mentioned……. “Here, we aim to discuss various types of innate immunity-mediated immunotherapy approaches focusing on hematological disorders, multiple myeloma, acute myelogenous leukemia, and solid tumor cells. Additionally, we aim to evaluate the role of iNKT and NK cells in antitumor responses, based on effector function, by highlighting key recent clinical immunotherapy findings.” However, there is no clear headings/subheadings related to this.

·       Author have included lot of content in section and subsections; however, these sections and subsections are not properly linked to each other. Thus, authors are suggested to link up the content.

·       Authors are suggested to add one/two table for describing the Immunological and clinical findings of iNKT/ NK-based immunotherapy.

·       Figures should be improved in terms of concept they are representing.

·       Authors should add the limitations of iNKT/ NK-based immunotherapy in conclusion section and also highlight the outcomes of this narrative review which will help in deciding the future therapeutics in disease management.

Typographical errors are present at several places throughout the manuscript. Authors must proofread the whole manuscript for possible errors.

Round 2

Reviewer 1 Report

The authors have carefully addressed all the comments. So, the manuscript should be accepted in its present form.

The authors could add a list of abbreviations.

Reviewer 3 Report

Dear Author,

I appreciate your efforts now it looks promising after Figure/Images.

Regards

PSG 

Reviewer 4 Report

Authors have revised the manuscript substantially thus it can now be considered for publication.